

**UV and Infrared Absorption Spectra, Atmospheric Lifetimes, and Ozone Depletion and**
**Global Warming Potentials for CCl₂FCCl₂F (CFC-112), CCl₃CClF₂ (CFC-112a), CCl₃CF₃**
**(CFC-113a), and CCl₂FCF₃ (CFC-114a)**
Maxine E. Davis,[1,2,3] François Bernard,[1,2] Max R. McGillen,[1,2]
Eric L. Fleming,[4,5] and James B. Burkholder[1]
[1] Earth System Research Laboratory, Chemical Sciences Division, National Oceanic and
Atmospheric Administration, Boulder, Colorado, USA.
[2] Cooperative Institute for Research in Environmental Sciences, University of Colorado,
Boulder, Colorado, USA.
[3] Michigan State University, Lyman Briggs College, East Lansing, MI.
[4] NASA Goddard Space Flight Center, Greenbelt, Maryland, USA.
[5] Science Systems and Applications, Inc., Lanham, Maryland, USA.
Corresponding author: James B. Burkholder, NOAA, 325 Broadway, Boulder, CO 80305, USA.
(James.B.Burkholder@noaa.gov)





**Abstract**  The potential impact of the recently observed $CCl_2FCCl_2F$ (CFC-112), $CCl_3CClF_2$
(CFC-112a), $CCl_3CF_3$ (CFC-113a), and $CCl_2FCF_3$ (CFC-114a) (chlorofluorocarbons, CFCs), on
stratospheric ozone and climate are presently not well characterized.  In this study, the UV
absorption spectra of these CFCs were measured between 192.5–235 nm over the temperature
range 207–323 K.  Precise parameterizations of the UV absorption spectra are presented.  A 2-D
atmospheric model was used to evaluate the CFC atmospheric loss processes, lifetimes, ozone
depletion potentials (ODPs), and the associated uncertainty ranges in these metrics.  The CFCs
are primarily removed in the stratosphere by short wavelength UV photolysis with calculated
global annually averaged steady-state lifetimes (years) of 63.6 (61.9–64.7), 51.5 (50.0–52.6),
55.4 (54.3–56.3), and 105.3 (102.9–107.4) for CFC-112, CFC-112a, CFC-113a, and CFC-114a,
respectively.  The range of lifetimes given in parentheses where obtained by including the 2σ
uncertainty in the UV absorption spectra and $O(^1D)$ rate coefficients in the model calculations.
The 2-D model was also used to calculate the CFC ozone depletion potentials (ODPs) with
values of 0.98, 0.86, 0.73, and 0.72 obtained for CFC-112, CFC-112a, CFC-113a, and CFC-
114a, respectively.  Using the infrared absorption spectra and lifetimes determined in this work,
the CFCs global warming potentials (GWPs) were estimated to be 4260 (CFC-112), 3330 (CFC-
112a), 3650 (CFC-113a), and 6510 (CFC-114a) for the 100-year time-horizon.







## 1. Introduction


Chlorofluorocarbons (CFCs) are potent ozone depleting and greenhouse gases that were
phased-out of production under the Montreal Protocol Agreement (1987) and its subsequent
amendments and adjustments. Laube et al. (2014) recently reported the first observation of
tetrachloro-1,2-difluoroethane ($CCl_2FCCl_2F$, CFC-112), tetrachloro-1,1-difluoroethane
($CCl_3CClF_2$, CFC-112a), and 1,1,1-trichloro-2,2,2-trifluoroethane ($CCl_3CF_3$, CFC-113a) in the
atmosphere with emission sources dating back to the 1960s. The atmospheric loading in the year
2000 were found to be ~0.5 ppt (CFC-112), ~0.08 ppt (CFC-112a), and ~0.3 ppt (CFC-113a),
which are minor compared to a total chlorine loading of 3.3 ppb (year 2012), where $CCl_3F$ (CFC-
11), $CCl_2F_2$ (CFC-12), and $CCl_2FCClF_2$ (CFC-113) account for ~60% of the total (WMO, 2014).
The atmospheric abundance of CFC-112 and CFC-112a was found to have leveled off in the late
1990's, while the abundance of CFC-113a was found to be increasing through to the present day,
which is contrary to the objectives of the Montreal Protocol. Laube et al. (2014) estimated the
stratospheric lifetimes for these substances, using a tracer-tracer analysis, to be 51 (37–82), 44
(28–98), and 51 (27–264) years for CFC-112, CFC-112a, and CFC-113a, respectively, where the
values in parentheses are the range of the lifetimes determined in their analysis. The inferred
ozone depletion potentials (ODPs) were 0.88 (0.62–1.44), 0.88 (0.5–2.19), and 0.68 (0.34–3.79)
for CFC-112, CFC-112a, and CFC-113a, respectively, where the range in parentheses was
derived from the range in the CFC lifetime given above. It is clear that the CFCs are long-lived
compounds and potent ozone depleting substances and greenhouse gases. It is expected that
these compounds would be predominantly removed from the atmosphere via short wavelength
UV photolysis, primarily in the stratosphere. However, to date, there are no UV absorption
spectra for these compounds available, which are needed to better evaluate their atmospheric
impact.
In this study, UV absorption spectra were measured for CFC-112, CFC-112a, CFC-113a,
and 1,1-dichlorotetrafluoroethane ($CCl_2FCF_3$, CFC-114a) between 192.5 and 235 nm over the
temperature range 207–323 K. The Goddard Space Flight Center (GSFC) 2-D atmospheric
model was used to evaluate the reactive and photolytic loss processes and calculate globally
averaged lifetimes and ozone depletion potentials. In addition, infrared absorption spectra were
measured at 296 K for these compounds and used to estimate their global warming potentials



(GWPs).  The present results are compared with results from the previous infrared studies of
Olliff and Fischer (1992; 1994) and Etminan et al. (2014) where possible.
**2. Experimental Details**
**2.1 UV Measurements**
The experimental apparatus has been described in detail previously (McGillen et al.,
2013; Papadimitriou et al., 2013a; 2013b) and is only briefly discussed here.  The output of a
stable 30 W deuterium ($D_2$) lamp light source was collimated and directed through a jacketed
90.4 ± 0.3 cm single pass absorption cell.  The beam exiting the cell was focused onto the
entrance slit (150 μm) of a 0.25 m monochromator and detected using a photomultiplier tube
(PMT).  The temperature of the absorption cell was controlled to within ±1 K.  Absorption
measurements were made at 10 discrete wavelengths at temperatures between 207 and 323 K.
Beer's law was applied to determine the absorption cross section, $\sigma(\lambda,T)$, at each
wavelength and temperature:
$$A(\lambda, T) = ln\left[\frac{I_0(\lambda)-I_d}{I(\lambda)-I_d}\right] = \sigma(\lambda, T) \times L \times [CFC] \qquad (1)$$
where $A(\lambda,T)$ is the absorbance at wavelength $\lambda$ and temperature T, $I_d$ is the signal recorded in
the absence of light, $I_0(\lambda)$ and $I(\lambda)$ are the measured signal in the absence and presence of the
CFC sample, L is the cell pathlength, and [CFC] is the gas-phase CFC concentration.  The PMT
signal was recorded with a 1 kHz sampling rate and a ~20 s average was used in the data
analysis.  $I_0(\lambda)$ was recorded at the beginning and end of each measurement, which typically
agreed to 0.1%, or better.  Absorbance measurements were made at each wavelength over a
range of CFC concentration under static conditions.  The CFCs were added to the absorption cell
from dilute mixtures and the CFC concentration was determined using the sample mixing ratio,
the absorption cell pressure and temperature, and the ideal gas law.  A linear least-squares fit of
$A(\lambda,T)$ versus [CFC] was used to obtain $\sigma(\lambda,T)$.
For the CFC-112 and CFC-112a measurements, an optical neutral density filter was
inserted between the $D_2$ lamp and the absorption cell to attenuate the probe beam and minimize
CFC loss due to photolysis (sample photolysis was not observed for CFC-113a and CFC-114a).
In addition, a mechanical shutter blocked the $D_2$ lamp beam while the absorption cell was being
filled.  Under most conditions, photolytic loss of the CFC-112 and CFC-112a was undetectable.
However, at the higher concentrations used in this study minor photolytic loss (<2%) was





observed.  In these cases, a least-squares fit of the first ~20 s of the PMT signal was used in the
data analysis to obtain the initial $I(\lambda)$ signal.

## 2.2 Infrared Absorption Measurements

Infrared absorption spectra at 296 K for CFC-112, CFC-112a, CFC-113a, and CFC-114a
were measured over the 500 to 4000 cm$^{-1}$ wavenumber range using Fourier transform infrared
(FTIR) spectroscopy.  Measurements were made using a 15 cm single pass absorption cell at a
resolution of 1 cm$^{-1}$ with 100 co-adds.  The CFC sample was introduced into the absorption cell
from a dilute mixture prepared off-line and the CFC concentration was determined using the
ideal gas law.  Absorption cross sections were determined using Beer's law, Eq. (1), with the
spectrum measurements consisting of ~10 different concentrations.  The concentration ranges
used were (in $10^{16}$ molecule cm$^{-3}$): (0.348–10.2), (0.453–3.84), (0.376–1.90), and (0.279–4.02)
for CFC-112, CFC-112a, CFC-113a, and CFC-114a, respectively.   The infrared absorption
spectra recorded for CFC-112 and CFC-112a were corrected for the presence of a minor (~4%)
isomer impurity as determined from a $^{19}$F NMR sample analysis.

## 2.3 Materials

Samples of $CCl_2FCCl_2F$ (CFC-112, 97% stated purity), $CCl_3CClF_2$ (CFC-112a, 96%
stated purity), $CCl_3CF_3$ (CFC-113a, 99% stated purity), and $CCl_2FCF_3$ (CFC-114a, 99.9% stated
purity) were obtained commercially.  The samples were processed in several freeze (77 K)-
pump-thaw cycles prior to use.  The CFC-114a sample was also treated with freeze (197 K)-
pump-thaw cycles to remove $CO_2$ from the sample.  The liquid CFC-112, CFC-112a, and CFC-
113a samples were stored under vacuum in Pyrex reservoirs.  The CFC-112 and CFC-112a
samples contained minor isomeric impurities, which were quantified using $^{19}$F NMR to be
0.960/0.040 (CFC-112a/CFC-112) for the CFC-112a sample and 0.963/0.0368 (CFC-112/CFC-
112a) for the CFC-112 sample.  Dilute mixtures of the CFCs in a He (UHP, 99.999%) bath gas
were prepared manometrically in 12 L Pyrex bulbs and used to deliver the CFC sample to the
UV and infrared absorption cells.  Over the course of the study, multiple gas mixtures were
prepared for each of the CFCs with mixing ratios ranging between 0.5 and 27%.  The dilute
mixtures were prepared with an estimated accuracy of ±~1%.  The UV and infrared spectra
obtained for the CFCs were independent of the sample mixing ratio and absorption cell total
pressure.  Pressures were measured using calibrated capacitance manometers.  Uncertainties
given throughout the paper are 2σ unless noted otherwise.



**3. Results and Discussion**

The absorption spectrum, $\sigma(\lambda,T)$, measurements obeyed Beer's law with fit precisions of

~1%, or less, for all wavelengths and temperatures included in this study.    Replicate
measurements using different sample mixing ratios, bath gas, range of absorption, and optical
filtering agreed to within the measurement precision and were combined in a global linear least-
squares fit in the final data analysis.

The UV absorption spectra of the CFC-112 and CFC-112a samples were measured at 10

discrete wavelengths between 192.5 nm and 235 nm at 5 discrete temperatures between 230 and
323 K.    The results, not corrected for the isomeric impurity present in the samples, are
summarized in Tables S1 and S2 and shown in Figures S1 and S2 of the Supporting Information.
To account for the isomeric impurity, $\sigma(\lambda,T)$ for CFC-112 and CFC-112a were parameterized
using the empirical formula:
$$\ln\left(\sigma(\lambda,T)\right) = \sum_i A_i \lambda_i{}^i + (T - 296) \sum_i B_i \lambda_i{}^i \tag{2}$$
The parameterizations reproduced the experimental data to better than ~2% over the wavelength
range most critical to atmospheric photolysis, i.e., between 195 and 215 nm.    The results from
the $^{19}$F NMR sample analysis were then used to obtain the final spectrum parameterizations.

The UV absorption spectra for CFC-113a and CFC-114a were measured at 10 discrete

wavelengths between 192.5 and 235 nm at 6 discrete temperatures between 207 and 323 K.    The
cross section results are given in Tables 1 and 2 and shown in Figures 1 and 2.    The CFC UV
absorption spectra were parameterized using Eq. (2).    The parameterizations reproduced the
experimental data to within ~4%, or better, as shown in Figures 1 and 2.

The fit parameters are given in Table 3 and a comparison of the parameterized 296 K

spectra is shown in Figure 3.    The UV absorption spectra of the CFCs are continuous over the
wavelength range included in this study with a precipitous decrease in cross section with
increasing wavelength.    A decrease in $\sigma(\lambda,T)$ with decreasing temperature was observed at
nearly all wavelengths included in this study with the temperature dependence being greatest at
the longer wavelengths, see Figures 1, 2, and S1 and S2.    The inclusion of the $\sigma(\lambda,323\ K)$
measurements, although not entirely atmospherically relevant, was included in the study to better
define the absorption spectrum temperature dependence and its parameterization.    As shown in
Figure 3, the UV absorption spectra for the CFCs show distinct differences in their absolute cross
sections and wavelength dependence over the region most critical for determining their



atmospheric photolysis rates, i.e., lifetimes. The spectra demonstrate that CFCs with increased
chlorine content are stronger absorbers in this wavelength region, although the molecular
structure of the molecule also plays an important role. For example, the $C_2Cl_4F_2$ isomer with
more chlorine atoms on a carbon atom, CFC-112a ($CCl_3CClF_2$), absorbs more strongly than
CFC-112 ($CCl_2FCCl_2F$).

The spectrum parameterizations given in Table 3 reproduce the experimental data very

well. The overall $2\sigma$ uncertainty in $\sigma(\lambda,T)$ for CFC-112, CFC-112a, CFC-113, and CFC-114a,
including estimated systematic errors, is estimated to be ~4% over the range of wavelengths and
temperatures included in this study.

The measured infrared spectra for each of the CFCs obeyed Beer's law with a fit

precision of ~0.3% and were independent of total pressure over the pressure range 20–250 Torr
(He bath gas). The infrared spectra are shown in Figure 4 and digitized spectra are available in
the Supporting Information. Table S3 in the Supporting Information provides a detailed
comparison of our results with those of Olliff and Fischer (1992; 1994) for all the CFCs and
Etminan et al. (2014) for CFC-113a. Overall the agreement between the studies is better than

10%.

**4. Atmospheric Implications**

The atmospheric loss processes, lifetimes, ODPs, and associated uncertainties for the

CFCs included in this study were quantified using the Goddard Space Flight Center (GSFC) 2-D
atmospheric model (Fleming et al., 2011). The calculations used the UV spectrum
parameterizations obtained in this work with an assumed unit photolysis quantum yield at all
wavelengths. As discussed in section 3, an overall $2\sigma$ uncertainty of 4% was used at all
wavelengths and temperatures for the UV cross sections of the four CFCs. For Lyman-
$\alpha$ (121.567 nm), absorption cross sections are not available for these CFCs and values (in units
of $10^{-17}$ $cm^2$ $molecule^{-1}$) of 13, 15, 9.8, and 2 were estimated for CFC-112, CFC-112a, CFC-
113a, and CFC-114a, respectively, based on values available for similar molecules (see Ko et al.
(2013), Chapter 3). An estimated Lyman-$\alpha$ cross section uncertainty factor of 2 ($2\sigma$) was used.
Rate coefficients for the O($^1$D) reaction with CFC-113a and CFC-114a were taken from
Baasandorj et al. (2011) with $2\sigma$ uncertainty factors of 1.25 and 1.2, respectively. Rate
coefficients for the O($^1$D) reaction with CFC-112 and CFC-112a were estimated to be $3 \times 10^{-10}$



cm$^3$ molecule$^{-1}$ s$^{-1}$ with a 0.9 reactive branching ratio and an uncertainty factor of 1.5 (2σ).  All
other kinetic and photochemical parameters were taken from Sander et al. (2011).  All model
results presented in this study are for year 2000 steady-state conditions.

Model calculations of the CFC fractional atmospheric loss processes are given in Table 4

and the altitude profiles for CFC-112 are shown in Figure 5.  The calculated atmospheric profiles
for CFC-112a, CFC-113a, and CFC-114a are provided in the Supporting Information.   UV
photolysis is the predominant atmospheric loss process for each of the CFCs.   Lyman-α
photolysis is important only in the mesosphere above 65 km; it has a negligible contribution to
the overall global loss (<0.001).  The O($^1$D) reaction is a minor stratospheric loss process, ~2%,
for CFC-112, CFC-112, and CFC-113a, but more significant for CFC-114a, ~7%.  The UV
photolysis and O($^1$D) reactive loss of the CFCs leads to the direct release of reactive chlorine and
the formation of chlorine containing radicals (Burkholder et al., 2015).

The CFC lifetimes were computed as the ratio of the annually averaged global

atmospheric burden to the vertically integrated annually averaged total global loss rate (Ko et al.,
2013).  The total global lifetime ($\tau_{Tot}$) was also separated by the troposphere ($\tau_{Trop}$, surface to the
tropopause, seasonally and latitude-dependent), stratosphere ($\tau_{Strat}$), and mesosphere ($\tau_{Meso}$, <1
hPa) using the total global atmospheric burden and the loss rate integrated over the different
atmospheric regions such that
$$\frac{1}{\tau_{Tot}} = \frac{1}{\tau_{Trop}} + \frac{1}{\tau_{Strat}} + \frac{1}{\tau_{Meso}} \qquad (2)$$
The 2-D model total global annually averaged lifetimes and the range in lifetimes are given in
Table 5.  The 2σ range in the lifetime was calculated using the absolute 2σ maximum and
minimum in the UV absorption spectra and estimated Lyman-α cross sections reported in the
present work, along with the 2σ uncertainties in the O($^1$D) rate coefficients taken from Sander et
al. (2011).   The CFCs are long-lived and primarily removed in the stratosphere by UV
photolysis. The uncertainty in the calculated lifetime due to the uncertainty in the UV absorption
spectra measured in this work is small, <2%. The absolute lifetime uncertainty due to the kinetic
and photochemical input parameters is expected to be small compared to that calculated using
different atmospheric models due to the individual model treatment of dynamics, chemistry,
radiation, numeric, and other processes (Chipperfield et al., 2014; Ko et al., 2013).





The model calculated stratospheric lifetimes for CFC-112, CFC-112a, and CFC-113a are
in reasonable agreement with the values of 51 (37–82), 44 (28–98), and 51 (27–264) years
reported by Laube et al. (2014) (uncertainty ranges in parentheses). The lifetimes reported by
Laube et al. were based on a tracer-tracer analysis (see Plumb and Ko (1992) and Volk et al.
(1997) for method details) using a reference CFC-11 lifetime of 45 years. Scaling to the 52 year
CFC-11 lifetime given in WMO (2014) brings the results into better agreement with the present
work. The range of lifetimes obtained in the model results, which was determined solely based
on the uncertainty in the kinetic and photochemical input parameters, is, however, significantly
less than obtained in the tracer-tracer analysis. It is worth noting that while the total global
lifetimes of the isomers CFC-112 and CFC-112a are similar, the lifetimes of CFC-113a (55.4
yrs) and CFC-114a (105.3 yrs) are substantially shorter (by ~60%) than those of the isomers
CFC-113 (93 yrs) and CFC-114 (189 yrs) (WMO, 2014).

**4.1. Ozone Depletion Potentials (ODPs)**

The semi-empirical and model calculated ODPs for the CFCs are given in Table 6. The
ODP was calculated following the methodology used previously (Fisher et al., 1990; Wuebbles,
1983). Steady-state simulations for year 2000 were run with the surface boundary conditions for
the four CFCs and CFC-11 (used as the reference compound) increased individually to obtain a
~1% depletion in annually averaged global total ozone. The ODP was then taken as the change
in global ozone per unit mass emission of the CFC relative to the change in global ozone per unit
mass emission of CFC-11. Each of these compounds is a potent ozone depleting substance. The
model calculated ODPs for CFC-112, CFC-112a, and CFC-113a are similar to the semi-
empirical values inferred by Laube et al. (2014). The small range (<±0.015) in the model ODP
values is primarily due to the relatively small uncertainty in the UV spectra obtained in this
work.
Table 6 also includes ODPs for CFC-113 and CFC-114. These are larger than the ODPs
for the isomers CFC-113a and CFC-114a (especially CFC-113 vs CFC-113a), likely due in part,
to the longer lifetimes of CFC-113 and CFC-114. For comparison with other related compounds,
the ODPs of CFC-115, CFC-12, and $CCl_4$ are also included in Table 6. This shows the general
decrease in ODP with decreasing chlorination among CFC-112a, CFC-112, CFC-113a, CFC-
113, CFC-114a, CFC-114, and CFC-115. We also note that the model ODPs for CFC-112 and
CFC-112a are generally similar, although slightly less, than $CCl_4$ which also contains 4 chlorine



atoms. For most of the compounds listed in Table 6, the model ODPs are larger than the semi-
empirical values, likely due in part, to differences in the observationally based fractional release
factors compared to the model calculations.
**4.2. Calculated Radiative Efficiencies (RE) and Global Warming Potentials (GWPs)**
Table 6 summarizes the radiative efficiencies (REs) for the CFCs calculated using the
methods described in Hodnebrog et al. (2013) and the global warming potentials (GWPs) for the
20, 100, and 500-year time-horizons using the lifetimes from this work. The CFCs are potent
greenhouse gases and radiative forcing agents due to their high REs and long atmospheric
lifetimes. The GWPs for these long-lived compounds are comparable, or less than, those of the
atmospherically most abundant CFCs, e.g. the 100 year time-horizon GWPs for CFC-11 ($CCl_3F$),
CFC-12 ($CCl_2F_2$), and CFC-113 ($CCl_2FCClF_2$) are 4660, 10200, and 5820, respectively (WMO,
2014).

**5. Conclusions**
Short wavelength UV absorption spectra for $CCl_2FCCl_2F$ (CFC-112), $CCl_3CClF_2$ (CFC-
112a), $CCl_3CF_3$ (CFC-113a), and $CCl_2FCF_3$ (CFC-114a) measured in this work between 192.5
and 235 nm and at temperatures in the range 207 to 323 K were combined with 2-D atmospheric
model calculations to assess their atmospheric loss processes, lifetimes, and ozone depletion
potentials (ODPs). Short wavelength UV photolysis was shown to be the predominant loss
process for the CFCs with global annually averaged lifetimes of 63.6, 51.5, 55.5, and 105.3
years, for CFC-112, CFC-112a, CFC-113a, and CFC-114a, respectively. The uncertainty in the
model-calculated lifetimes due primarily to the $2\sigma$ uncertainty in the UV absorption spectra
reported in this work, was found to be small, <3%. These CFCs are potent ozone depleting
substances with 2-D model calculated ODPs of 0.98, 0.86, 0.73, and 0.72 for CFC-112, CFC-
112a, CFC-113a, and CFC-114a, respectively. The uncertainty in the model calculated ODPs
due to the uncertainty in the UV spectra and $O(^1D)$ reactive loss is small, <±0.015. These CFCs
are also potent greenhouse gases with GWPs comparable to those of the most abundant CFCs
present in the atmosphere.
**Acknowledgments**. This work was supported in part by NOAA's Atmospheric Chemistry,
Carbon Cycle, and Climate (AC4) Program and NASA's Atmospheric Composition Program.
Supporting information includes digitized infrared spectra as well as additional figures, model
results, and tables.





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



**Table 1.** CCl$_3$CF$_3$ (CFC-113a) UV Absorption Cross Section Data ($10^{-20}$ cm$^2$ molecule$^{-1}$, base e)
Obtained in This Work.

| λ (nm) | 323 K | 296 K | 271 K | 250 K | 232 K | 207 K |
|---|---|---|---|---|---|---|
| 192.5 | 131.6 ± 1.5 | 132.5 ± 1.1 | 136.9 ± 1.0 | 137.2 ± 0.9 | 141.4 ± 1.6 | 139.7 ± 0.9 |
| 195 | 103.9 ± 0.2 | 106.6 ± 0.6 | 106.8 ± 0.9 | 107.5 ± 1.2 | 110.2 ± 1.0 | 111.0 ± 0.3 |
| 200 | 64.3 ± 0.2 | 63.9 ± 0.6 | 63.9 ± 1.2 | 63.6 ± 0.6 | 64.5 ± 0.6 | 63.5 ± 0.4 |
| 205 | 35.3 ± 0.14 | 34.1 ± 0.2 | 33.5 ± 0.13 | 33.2 ± 0.2 | 31.9 ± 0.3 | 31.3 ± 0.3 |
| 210 | 17.3 ± 0.10 | 16.2 ± 0.1 | 15.3 ± 0.1 | 14.4 ± 0.1 | 13.9 ± 0.17 | 12.5 ± 0.2 |
| 215 | 7.99 ± 0.02 | 7.25 ± 0.01 | 6.58 ± 0.02 | 5.94 ± 0.06 | 5.77 ± 0.06 | 5.26 ± 0.07 |
| 220 | 3.57 ± 0.014 | 3.07 ± 0.02 | 2.65 ± 0.007 | 2.36 ± 0.02 | 2.23 ± 0.008 | 2.09 ± 0.02 |
| 225 | 1.55 ± 0.014 | 1.29 ± 0.01 | 1.04 ± 0.004 | 0.912 ± 0.006 | 0.813 ± 0.01 | 0.778 ± 0.04 |
| 230 | 0.673 ± 0.009 | 0.521 ± 0.004 | 0.418 ± 0.003 | 0.357 ± 0.008 | 0.322 ± 0.003 | |
| 235 | 0.297 ± 0.018 | 0.208 ± 0.001 | 0.157 ± 0.006 | 0.139 ± 0.006 | | |

* Quoted uncertainties are 2σ fit precision values (rounded off).




**Table 2.** $CCl_2FCF_3$ (CFC-114a) UV Absorption Cross Section Data ($10^{-20}$ $cm^2$ molecule$^{-1}$, base e) Obtained in This Work.

| λ (nm) | 323 K | 296 K | 271 K | 250 K | 232 K | 207 K |
|---|---|---|---|---|---|---|
| 192.5 | 32.8 ± 0.2 | 32.2 ± 0.3 | 31.6 ± 0.2 | 30.7 ± 0.2 | 30.0 ± 0.3 | 28.2 ± 0.2 |
| 195 | 21.8 ± 0.1 | 20.7 ± 0.1 | 19.9 ± 0.1 | 19.0 ± 0.1 | 18.4 ± 0.1 | 17.3 ± 0.1 |
| 200 | 8.72 ± 0.01 | 7.86 ± 0.045 | 7.26 ± 0.02 | 6.70 ± 0.03 | 6.26 ± 0.04 | 5.88 ± 0.05 |
| 205 | 3.31 ± 0.01 | 2.86 ± 0.01 | 2.50 ± 0.03 | 2.29 ± 0.02 | 2.12 ± 0.02 | 1.91 ± 0.02 |
| 210 | 1.21 ± 0.003 | 0.991 ± 0.003 | 0.835 ± 0.006 | 0.757 ± 0.006 | 0.655 ± 0.083 | 0.555 ± 0.002 |
| 215 | 0.440 ± 0.002 | 0.345 ± 0.001 | 0.276 ± 0.001 | 0.246 ± 0.006 | 0.197 ± 0.001 | 0.168 ± 0.001 |
| 220 | 0.162 ± 0.002 | 0.118 ± 0.0004 | 0.0926 ± 0.0003 | 0.0786 ± 0.0013 | 0.0626 ± 0.0003 | 0.0534 ± 0.0014 |
| 225 | 0.0600 ± 0.001 | 0.0409 ± 0.0006 | 0.0307 ± 0.0002 | 0.0253 ± 0.0002 | 0.0204 ± 0.0004 | 0.0176 ± 0.0046 |
| 230 | | 0.0147 ± 0.0004 | 0.0110 ± 0.0002 | | | |
| 235 | | 0.00553 ± 0.00025 | | | | |

* Quoted uncertainties are 2σ fit precision values (rounded off).





**Table 3.** Parameterization of the UV absorption spectra for $CCl_2FCCl_2F$ (CFC-112), $CCl_3CClF_2$
(CFC-112a), $CCl_3CF_3$ (CFC-113a), and $CCl_2FCF_3$ (CFC-114a) obtained in this work. The
parameterization is for wavelengths between 192.5 to 235 nm and temperatures between 230 and
323 K for CFC-112 and CFC-112a and between 207 and 323 K for CFC-113a and CFC-114a.
Units: $\sigma(\lambda,T)$ ($cm^2$ molecule$^{-1}$, base e), $\lambda$ (nm), and T (K)

$$\ln(\sigma(\lambda, T)) = \sum_i A_i \lambda_i{}^i + (T - 296) \sum_i B_i \lambda_i{}^i$$

| Molecule | $i$ | $A_i$ | $B_i$ |
|---|---|---|---|
| $CCl_2FCCl_2F$ (CFC-112) | | | |
| | 0 | -1488.6207 | 6.04688 |
| | 1 | 18.43604 | -0.0801501 |
| | 2 | -0.02897393 | 0.0001201698 |
| | 3 | -0.00051504703 | $2.610366 \times 10^{-6}$ |
| | 4 | $2.644261 \times 10^{-6}$ | $-1.3959106 \times 10^{-8}$ |
| | 5 | $-3.7258313 \times 10^{-9}$ | $2.0719264 \times 10^{-11}$ |
| $CCl_3CClF_2$ (CFC-112a) | | | |
| | 0 | -560.3404 | 10.37492 |
| | 1 | 9.534427 | -0.182485408 |
| | 2 | -0.06987945 | 0.0011614979 |
| | 3 | 0.0002657157 | $-2.9864183 \times 10^{-6}$ |
| | 4 | $-5.491224 \times 10^{-7}$ | $1.547878 \times 10^{-9}$ |
| | 5 | $4.993769 \times 10^{-10}$ | $3.36518 \times 10^{-12}$ |
| $CCl_3CF_3$ (CFC-113a) | | | |
| | 0 | -319.173 | 2.89174 |
| | 1 | 2.70954 | -0.0348043 |
| | 2 | 0.00457404 | $3.6233 \times 10^{-5}$ |
| | 3 | -0.0001288147 | $1.08853 \times 10^{-6}$ |
| | 4 | $4.71409 \times 10^{-7}$ | $-5.25744 \times 10^{-9}$ |
| | 5 | $-5.35388 \times 10^{-10}$ | $7.26095 \times 10^{-12}$ |
| $CCl_2FCF_3$ (CFC-114a) | | | |
| | 0 | -253.6338 | 0.52031 |
| | 1 | 2.899454 | -0.005044 |
| | 2 | -0.0081158 | $1.6142 \times 10^{-6}$ |
| | 3 | $-3.68328 \times 10^{-5}$ | $7.2259 \times 10^{-8}$ |
| | 4 | $2.071842 \times 10^{-7}$ | $2.4996 \times 10^{-11}$ |
| | 5 | $-2.5764 \times 10^{-10}$ | $-5.9642 \times 10^{-13}$ |

379

380



381

**Table 4.** Fractional losses and ranges (in parenthesis) for CCl$_2$FCCl$_2$F (CFC-112), CCl$_3$CClF$_2$ (CFC-112a), CCl$_3$CF$_3$ (CFC-113a), and CCl$_2$FCF$_3$ (CFC-114a) calculated using the GSFC 2-D model and the UV absorption spectra and estimated Lyman-α cross sections reported in this work

| Molecule | Lyman-α | 190-230 nm | O($^1$D) |
|---|---|---|---|
| CCl$_2$FCCl$_2$F (CFC-112) | <0.001 | 0.978 (0.953–0.99) | 0.022 (0.047–0.01) |
| CCl$_3$CClF$_2$ (CFC-112a) | <0.001 | 0.979 (0.955–0.99) | 0.021 (0.045–0.01) |
| CCl$_3$CF$_3$ (CFC-113a) | <0.001 | 0.979 (0.968–0.986) | 0.021 (0.032–0.014) |
| CCl$_2$FCF$_3$ (CFC-114a) | <0.001 | 0.929 (0.903–0.948) | 0.071 (0.097–0.052) |








**Table 5.** Atmospheric lifetimes $(\tau)$[a] and ranges[b] (years) for $CCl_2FCCl_2F$ (CFC-112), $CCl_3CClF_2$
(CFC-112a), $CCl_3CF_3$ (CFC-113a), and $CCl_2FCF_3$ (CFC-114a) calculated using the GSFC 2-D
model and the UV absorption spectra reported in this work

| Molecule | Tropospheric | | Stratospheric | | Mesospheric | Total | |
|---|---|---|---|---|---|---|---|
| | $\tau$ | $\tau$ Range | $\tau$ | $\tau$ Range | $\tau$ | $\tau$ | $\tau$ Range |
| $CCl_2FCCl_2F$ (CFC-112) | 2276 | (1718–2710) | 65.4 | (64.2–66.3) | $>10^6$ | 63.6 | (61.9–64.7) |
| $CCl_3CClF_2$ (CFC-112a) | 1187 | (938–1371) | 53.8 | (52.8–54.6) | $>10^6$ | 51.5 | (50.0–52.6) |
| $CCl_3CF_3$ (CFC-113a) | 1476 | (1290–1645) | 57.5 | (56.7–58.3) | $>10^6$ | 55.4 | (54.3–56.3) |
| $CCl_2FCF_3$ (CFC-114a) | 8312 | (6286–10480) | 106.7 | (104.7–108.6) | $3 \times 10^5$ | 105.3 | (102.9–107.4) |

[a] Global annually averaged values;  [b] Calculated using $2\sigma$ upper and lower limits of the UV
absorption cross sections and estimated Lyman-$\alpha$ cross sections reported in this work (see text)
and $O(^1D)$ rate coefficient uncertainties from Sander et al. (2011).




**Table 6.** Lifetimes, ozone depletion potentials (ODPs), radiative efficiencies (RE), and global
warming potentials (GWPs) obtained in this work and literature values for comparison

| Molecule | Lifetime (years) | Ozone Depletion Potential (ODP) | | Radiative Efficiency ($W\ m^{-2}\ ppb^{-1}$) | Global Warming Potential Time Horizons (years) | | |
|---|---|---|---|---|---|---|---|
| | | semi-empirical | 2-D Model [d] | | 20 | 100 | 500 |
| $CCl_2FCCl_2F$ (CFC-112) | 63.6 | 0.88 (0.62-1.44) [a] | 0.98 (±0.015) | 0.28 | 5330 | 4260 | 1530 |
| $CCl_3CClF_2$ (CFC-112a) | 51.5 | 0.88 (0.50-2.19) [a] | 0.86 (±0.015) | 0.25 | 4600 | 3330 | 1110 |
| $CCl_3CF_3$ (CFC-113a) | 55.4 | 0.68 (0.34-3.79) [a] | 0.73 (±0.01) | 0.24 | 4860 | 3650 | 1240 |
| $CCl_2FCF_3$ (CFC-114a) | 105.3 | | 0.72 (±0.01) | 0.28 | 6750 | 6510 | 3000 |
| $CCl_2FCClF_2$ (CFC-113) | 93 [b] | 0.81-0.82 [b] | 0.95 | 0.30 [b] | 6490 [b] | 5820 [b] | |
| $CClF_2CClF_2$ (CFC-114) | 189 [b] | 0.50 [b] | 0.78 | 0.31 [b] | 7710 [b] | 8590 [b] | |
| $CClF_2CF_3$ (CFC-115) | 540 [b] | 0.26 [b] | 0.44 | 0.20 [b] | 5860 [b] | 7670 [b] | |
| $CCl_2F_2$ (CFC-12) | 102 [b] | 0.73-0.81 [b] | 1.01 | 0.32 [b] | 10800 [b] | 10200 [b] | |
| $CCl_4$ | 26 [b,c] | 0.72 [b] | 1.06 | 0.17 [b] | 3480 [b] | 1730 [b] | |

[a] Semi-empirical ODPs and uncertainty ranges taken from Laube et al. (2014).
[b] Taken from WMO (2014).
[c] $CCl_4$ stratospheric lifetime of 44 years given in WMO (2014).
[d] The uncertainty range in the model calculated ODPs is due solely to the uncertainty in the UV
and Lyman-$\alpha$ (estimated) spectra obtained in this work and uncertainty in the $O(^1D)$ rate
coefficients taken from Sander et al. (2011).






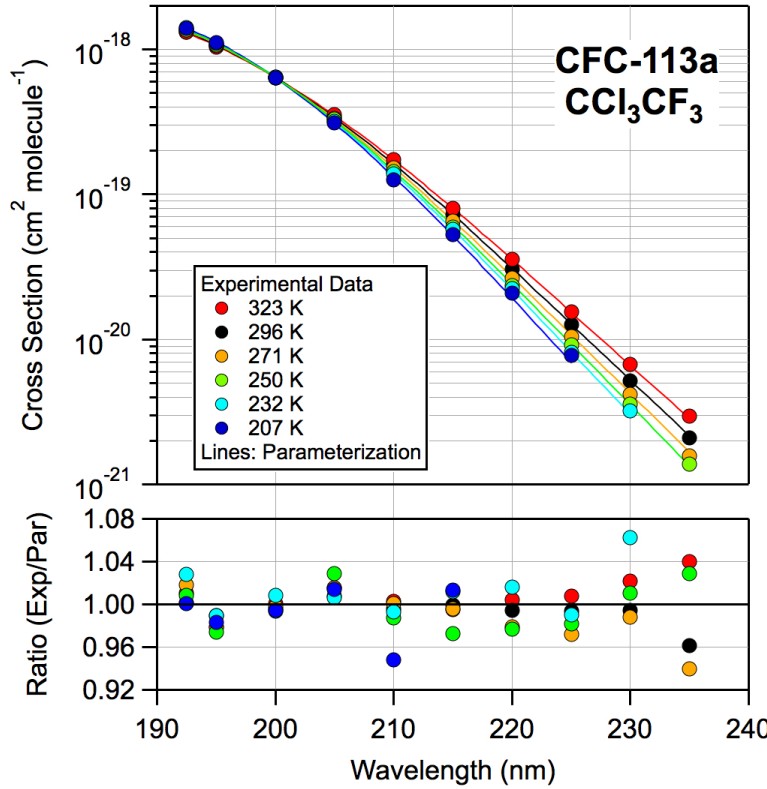


**Figure 1**. $CCl_3CF_3$ (CFC-113a) UV absorption spectrum (base e) and parameterization obtained in this work. Cross section data (symbols, Table 1) and the parameterization of the data using the empirical formula and parameters given in Table 3 (see text). The lower frame shows the overall quality of the parameterization.







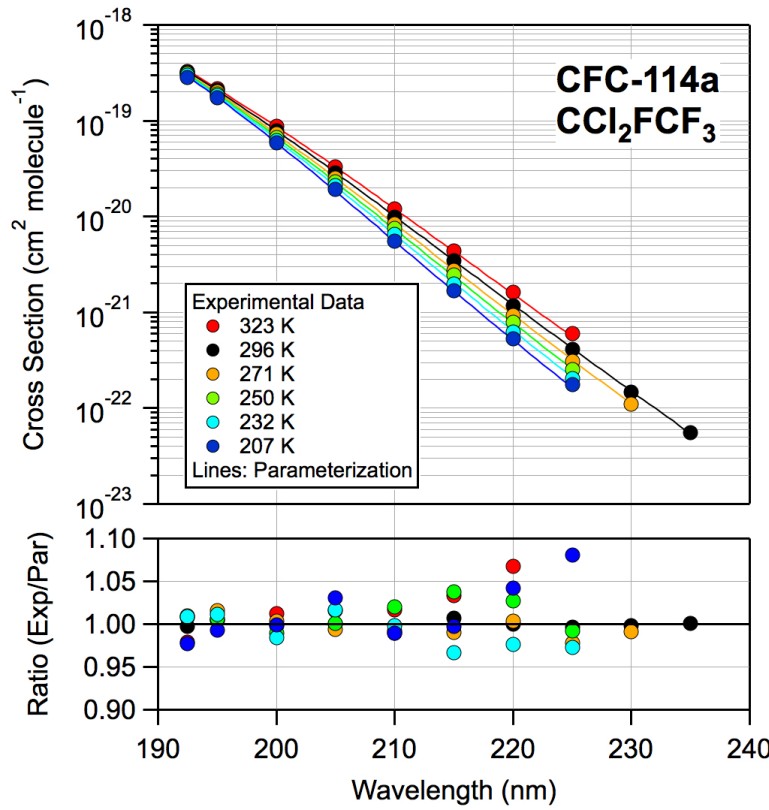


**Figure 2**. $CCl_2FCF_3$ (CFC-114a) UV absorption spectrum (base e) and parameterization
obtained in this work. Cross section data (symbols, Table 2) and the parameterization of the data
using the empirical formula and parameters given in Table 3 (see text). The lower frame shows
the overall quality of the parameterization.






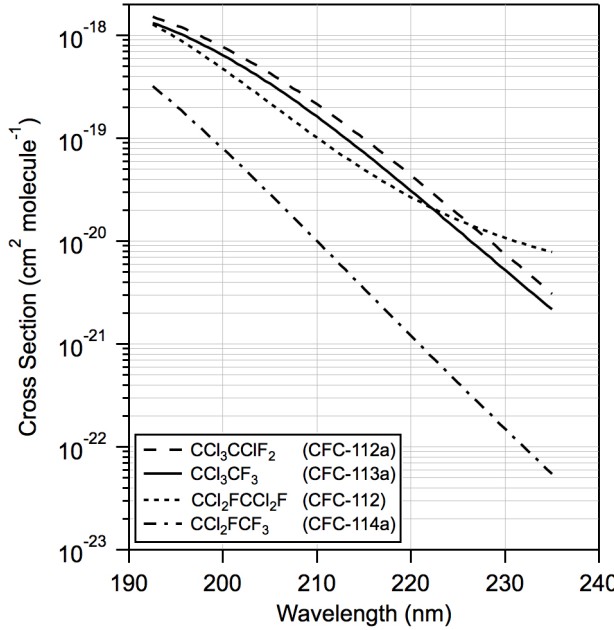


**Figure 3.** UV absorption spectra (base e) of CFC-112, CFC-112a, CFC-113a, and CFC-114a at
296 K calculated using the parameterization from this work, Table 3, over the wavelength range
of our experimental measurements.





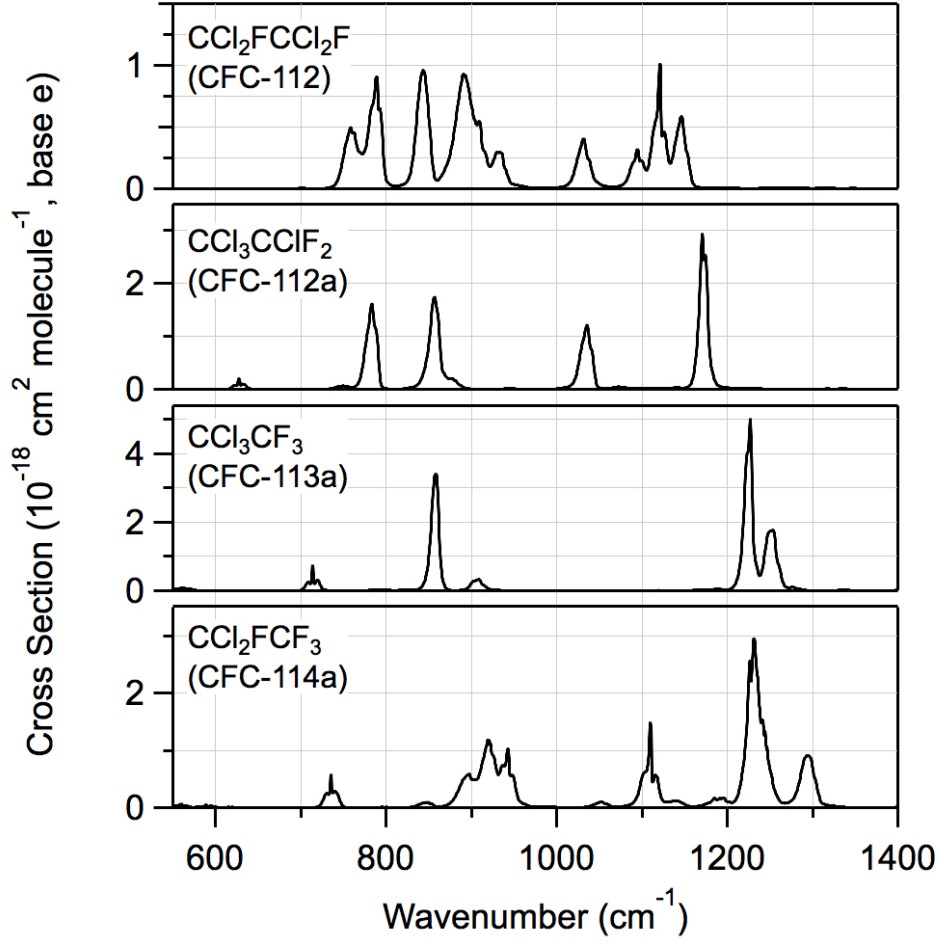

**Figure 4.** Infrared absorption spectra of CCl$_2$FCCl$_2$F (CFC-112), CCl$_3$CClF$_2$ (CFC-112a),
CCl$_3$CF$_3$ (CFC-113a), and CCl$_2$FCF$_3$ (CFC-114a) at 296 K obtained in this work.






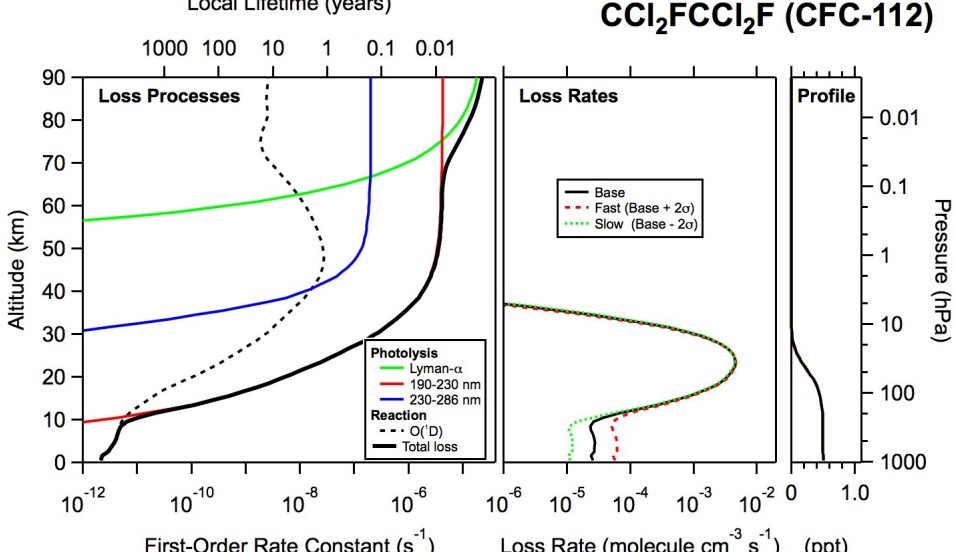

**Figure 5.** Global annually averaged vertical profiles of the atmospheric loss processes,
molecular loss rates, and mixing ratio for CCl$_2$FCCl$_2$F (CFC-112) calculated using the GSFC 2-
D atmospheric model for year 2000. The model calculations were performed using the CFC-112
UV absorption spectrum from this work and other model input parameters taken from the
literature as described in the text. The global annually averaged lifetime for CFC-112 was
calculated to be 63.6 (61.9–64.7) years.