# Peer review of "UV and Infrared Absorption Spectra, Atmospheric Lifetimes, and Ozone Depletion and Global Warming Potentials for CCl2FCCl2F (CFC-112), CCl3CClF2 (CFC-112a), CCl3CF3 (CFC-113a), and CCl2FCF3 (CFC-114a) Maxine E. D"

_Atmospheric Chemistry and Physics, 2016_

## Referee Comment (RC1) · Anonymous Referee #1 · 15 Mar 2016

The paper by Davis et al. combines measurements of UV and IR absorption cross sections of 4 chlororfluorocarbons (CFC) with 2D-modelling to derive atmospheric lifetimes, ozone depletion poltentials and GWP values for these species. Three of the CFCs have recently been reported to be present in the atmosphere at low concentrations. The paper is will written, the methodology is sound, the assumptions and conclusions are well justified and the scope of the paper is clearly relevant for ACP. I only have one major and a few minor suggestions for the authors, which I suggest they

should consider before publication.

Major point:

I find the uncertainty estimate unrealistic. The uncertainty which is reported here is purely the uncertainty due to kinetical and photochemical data. Not the uncertainty of the derived atmospheric lifetime. One important point in this respect is how fast tropospheric air is transported to the stratospheric loss region. The model lifetime will thus depend strongly on model transport. I therefore think it is unrealistic to assume that the model can really constrain the atmospheric lifetime this closely. For this, a thorough investigation of model transport would be necessary. I suggest that the authors discuss this point more closely and that they include a discussion on the uncertainty of the atmospheric lifetime due to transport. I further suggest that knowledge of actinic fluxes and the underlying uncertainties should be discussed in the uncertainty estimate.

Minor suggestions:

Introduction: I suggest stating more clearly that only three of the species investigated here have recently been observed and make a statement on whether there are indications of a presence of CFC114a in the atmosphere.

p.5.l 130: on what is this estimate based?

p.6.l. 140: how wide are the individual wavelength bands?

p.7.l. 193: please indicate if the error estimates are from the paper by Bassandorj.

p.8.l219 ff.: to what extend does the uncertainty in actinic flux influence the lifetimes and their uncertainties? See also major point above.

Conclusion: I suggest adding a short statement on the concentrations in the atmosphere and the global importance of these species.

---

## Referee Comment (RC2) · Anonymous Referee #2 · 1 Apr 2016

This is a very good manuscript reporting UV and IR spectra of four previously under-studied chlorofluorocarbons in the atmosphere as well as deriving relevant properties with regard to global warming and stratospheric ozone depletion. I recommend it for publication once the points below have been addressed, in particular the two major concerns on the discussion of IR data and on model uncertainties.

L49: Should be "atmospheric loadings". Besides, given in the following are actually not loadings but mole fractions.

L50-51: ppt and ppb not explained

L73-74 What is missing from the introduction is an overview of the current literature on the IR spectra of these CFCs (and in particular their shortcomings), such as the papers mentioned in these two lines.

L82-83: Why was this range chosen?

L109: Please define "co-adds". Also, the make of the FTIR, the cell material and the detector type are not given.

L144-49 It needs to be made clearer how equation 2 was used to correct for isomeric impurities, especially since that same equation is later on used for CFC-113a and -114a.

L161 It would be useful to explain to the reader which temperatures are "atmospherically relevant" and why.

L179-180 No discussion of IR spectra in any detail. For instance, which features of the spectra can be assigned to certain functional groups? And which agree best with other recorded spectra? In which spectral region are the biggest differences and what could be causing this?

L198-206 Given that there is a published data set of observed stratospheric mole fractions, which the authors refer to repeatedly I am surprised that no comparison between measurements and model have been attempted at all.

L209-210 Which definitions were used to define those regions?

L215-218 It seems very surprising that the atmospheric model should introduce no uncertainty at all. This is probably the main reason why the uncertainty ranges in the lifetimes given in Table 5 are so small, and in fact probably too small. One idea how to approach this problem would be to compare the loss rates derived by this model with observations for other more well-known molecules with similar loss distributions.

L245-247 This is related to my previous comment. The small range is not caused by the small uncertainty in the UV spectra but due to not including the probably substantially larger model uncertainties. This creates the impression that the lifetimes and ODPs estimated here are far superior to previous work; which they might be, but this is currently not proven.

L256-257 The term "fractional release factors" is explained nowhere in this manuscript. Why would they make a difference to the ODPs?

L258-266 Again, the IR spectra derived in this work are not discussed at all. In this case it is not even mentioned that they were used in this calculation. Also, a comparison with published REs and GWPs would be useful here as well as in Table 6.

---

## Author Comment (AC1) · 31 May 2016

We thank the referee for their constructive comments.

Response to Anonymous Referee #1

I find the uncertainty estimate unrealistic. The uncertainty which is reported here is purely the uncertainty due to kinetical and photochemical data. Not the uncertainty of the derived atmospheric lifetime. One important point in this respect is how fast tropospheric air is transported to the stratospheric loss region. The model lifetime will thus depend strongly on model transport. I therefore think it is unrealistic to assume that the model can really constrain the atmospheric lifetime this closely. For this, a thorough investigation of model transport would be necessary. I suggest that the authors discuss this point more closely and that they include a discussion on the uncertainty of the atmospheric lifetime due to transport. I further suggest that knowledge of actinic fluxes and the underlying uncertainties should be discussed in the uncertainty estimate.

Author Response: We certainly agree that there is uncertainty in the total derived lifetime due to the model transport uncertainty. However, investigation of this issue would require, for example, multiple sensitivity simulations with varying transport rates used in the same model, or base simulations from multiple models (which have different transport rates). The impacts of this transport uncertainty would then have to be evaluated against long-lived tracer observations, e.g. as done in SPARC Lifetime report, 2013, Chapter 5. However, such an extensive transport evaluation is far outside the scope of the present paper.

As for the actinic fluxes, there is uncertainty, for example, in the J[O2] and J[O3] cross sections. But again, addressing this issue thoroughly is beyond the scope of this paper.

In our study, we address only the uncertainty associated with the kinetic and photochemical data and its impact on the total lifetime. For this evaluation, we used a well-established and vetted 2-D atmospheric model that enables us to perform multiple model runs with different kinetic and photochemical input parameters at a reasonable cost. We do not attempt to address all of the processes that would contribute to lifetime uncertainty. This approach, which has been used previously, is used to get a better handle on whether the uncertainty in the lifetime is due to the kinetic and photochemical input parameters, i.e., laboratory data, or the details specific to a given model. Our results show that the uncertainty in the laboratory data is relatively small for these molecules and probably much smaller than the lifetime variability obtained using different models, e.g. the SPARC (2013) lifetime report found lifetime differences of 10-15%

between different models. That is, the absolute lifetimes obtained from models might differ from our 2-D model calculations, but the majority of the difference is not due to the kinetic and photochemical input parameters.

Action Taken: Some of the confusion on this issue is due to not stating these points clearly in the text. To address both referees' comments on this issue, we have re-worded the text in the Abstract, the Atmospheric Implications section, Table 6, and the Conclusions to specifically state that the reported uncertainty ranges in the lifetimes and ODPs are due to the kinetic and photochemical uncertainty. We have removed wording such as "the uncertainty is primarily due to . . ." since this can be interpreted as though the other sources of uncertainty are unimportant (e.g. transport, actinic fluxes).

Minor suggestions:

Referee Comment: Introduction: I suggest stating more clearly that only three of the species investigated here have recently been observed and make a statement on whether there are indications of a presence of CFC114a in the atmosphere.

Author Response: We can make this point more clearly in the text.

Action Taken: First sentence in Abstract: "The potential impact of the recently observed CCl2FCCl2F (CFC-112), CCl3CClF2 (CFC-112a), CCl3CF3 (CFC-113a), and CCl2FCF3 (CFC-114a) (chlorofluorocarbons, CFCs), on stratospheric ozone and climate are presently not well characterized." revised to ""The potential impact of CCl2FCF3 (CFC-114a) and the recently observed CCl2FCCl2F (CFC-112), CCl3CClF2 (CFC-112a), and CCl3CF3 (CFC-113a) chlorofluorocarbons (CFCs) on stratospheric ozone and climate are presently not well characterized.". Introduction: Inserted the following sentence: "Atmospheric measurements of CFC-114 are estimated to include a ∼10% fraction due to CFC-114a (WMO, 2014). The atmospheric lifetime of CFC-114a is estimated to be similar to that of CFC-12, i.e., ∼100 years (WMO, 2014)".

Referee Comment: p.5.l 130: on what is this estimate based?

Author Response: The estimated uncertainty in the dilute mixture mixing ratios was based on the accuracy of the absolute pressure measurements. Due to the fact that numerous mixtures were used over the course of this study only an estimated uncertainty, which is relatively small and does not make a significant contribution to the overall cross section uncertainty, is given here.

Action Taken: The text has been revised as follows: "The dilute mixtures were prepared with an estimated accuracy of $\pm\sim$1%." was revised to "The dilute mixtures were prepared with an estimated accuracy of $\pm\sim$1% (based on the estimated pressure measurement uncertainty).".

Referee Comment: p.6.l. 140: how wide are the individual wavelength bands?

Author Response: I believe that the reviewer is asking for the resolution of the UV absorption measurement. The resolution was $\sim$1 nm.

Action Taken: The text in the first paragraph of section 2.1 was revised as follows: "The beam exiting the cell was focused onto the entrance slit (150 um) of a 0.25 m monochromator ($\sim$1 nm resolution) and detected using a photomultiplier tube (PMT).".

Referee Comment: p.7.l. 193: please indicate if the error estimates are from the paper by Bassandorj.

Author Response: The error estimates were taken from the preliminary 2015 JPL data evaluation, which is now publicly available.

Action Taken: A reference to the 2015 data evaluation has been added to the text for clarification.

Referee Comment: p.8.l219 ff.: to what extend does the uncertainty in actinic flux influence the lifetimes and their uncertainties? See also major point above.

Author Response: See response to first comment.

[Figure]

Action Taken: See response to first comment.

Referee Comment: Conclusion: I suggest adding a short statement on the concentrations in the atmosphere and the global importance of these species.

Author Response: The atmospheric concentrations of the compounds included in this work is presented in the Introduction with the appropriate references. These compounds make a minor contribution to the total chlorine in the atmosphere, as pointed out in the Introduction, but it is important that the fate of these species are characterized by laboratory studies.

Action Taken: None

---

## Author Comment (AC2) · 31 May 2016

We thank the referee for their constructive comments.

Response to Anonymous Referee #2

This is a very good manuscript reporting UV and IR spectra of four previously under-studied chlorofluorocarbons in the atmosphere as well as deriving relevant properties with regard to global warming and stratospheric ozone depletion. I recommend it for

publication once the points below have been addressed, in particular the two major concerns on the discussion of IR data and on model uncertainties.

Referee Comment: L49: Should be "atmospheric loadings". Besides, given in the following are actually not loadings but mole fractions.

Author Response: Okay

Action Taken: In the first paragraph of the Introduction we have changed "atmospheric loading" and "atmospheric abundance" to "atmospheric mixing ratio".

Referee Comment: L50-51: ppt and ppb not explained

Author Response: Okay

Action Taken: We have included "(part per trillion)" and "(part per billion)" after the first appearance of ppt and ppb, respectively.

Referee Comment: L73-74 What is missing from the introduction is an overview of the current literature on the IR spectra of these CFCs (and in particular their shortcomings), such as the papers mentioned in these two lines.

Author Response: Such a review is not necessary at this stage of the paper. However, we could clarify which molecules were included in the Olliff and Fischer and Etminan et al. studies.

Action Taken: We have revised the text as follows: "... previous infrared studies of Olliff and Fischer (1992; 1994) (CFCs 112, 112a, 113a, and 114a) and Etminan et al. (2014) (CFC-113a) where possible.

Referee Comment: L82-83: Why was this range chosen?

Author Response: The temperature range of the spectrum measurements was chosen in an attempt to represent stratospheric temperatures where these compounds would photolyze. The elevated temperature was included to improve the spectra parameterizations, as stated in the text.

Action Taken: The text was revised as follows: "Absorption measurements were made at 10 discrete wavelengths at temperatures between 207 and 323 K to enable spectrum parameterizations appropriate for stratospheric conditions.".

Referee Comment: L109: Please define "co-adds". Also, the make of the FTIR, the cell material and the detector type are not given.

Author Response: "co-adds" is a standard Fourier transform spectroscopy terminology that does not require definition. We do not endorse manufactures in our work and therefore do not include make and model of commercial instruments. We should, however, identify the material of the cell and detector.

Action Taken: The text in this section has been revised as follows: "Measurements were made using a 15 cm single pass Pyrex absorption cell and a MCT detector at a resolution of 1 cm-1 with 100 co-adds. ".

Referee Comment: L144-49 It needs to be made clearer how equation 2 was used to correct for isomeric impurities, especially since that same equation is later on used for CFC-113a and -114a.

Author Response: There is nothing too complicated done here. It is basically solving two linear equations for two unknowns.

Action Taken: None

Referee Comment: L161 It would be useful to explain to the reader which temperatures are "atmospherically relevant" and why.

Author Response: This was addressed in our revisions above (L82 comment).

Action Taken: None

Referee Comment: L179-180 No discussion of IR spectra in any detail. For instance,

which features of the spectra can be assigned to certain functional groups? And which agree best with other recorded spectra? In which spectral region are the biggest differences and what could be causing this?

Author Response: A detailed comparison of the present results with the available previous work is provided in the supplement. The agreement with the Olliff and Fischer studies and the Etminan et al. study for CFC-113a is relatively good. Therefore, there is not a need for too much discussion. A discussion of the fundamental infrared spectroscopy of CFCs might be of interest to some readers, but was not the focus of the present work.

Action Taken: None

Referee Comment: L198-206 Given that there is a published data set of observed stratospheric mole fractions, which the authors refer to repeatedly I am surprised that no comparison between measurements and model have been attempted at all.

Author Response: The observations are for the surface concentrations only, and we have used these as surface mixing ratio boundary conditions input into the model. However, to our knowledge there are no observations of the vertical profiles of these newly-detected compounds to compare with the model.

Action Taken: We have added text in section 4 to state that the model uses the observed surface concentrations as input boundary conditions.

Referee Comment: L209-210 Which definitions were used to define those regions?

Author Response: Our definitions of the troposphere, stratosphere, and mesosphere are stated in the text as is and are also defined in the referenced Ko et al. SPARC (2013) lifetime report and Fleming et al. (2011) paper.

Action Taken: None

Referee Comment: L215-218 It seems very surprising that the atmospheric model

should introduce no uncertainty at all. This is probably the main reason why the uncertainty ranges in the lifetimes given in Table 5 are so small, and in fact probably too small. One idea how to approach this problem would be to compare the loss rates derived by this model with observations for other more well-known molecules with similar loss distributions. Referee Comment: L245-247 This is related to my previous comment. The small range is not caused by the small uncertainty in the UV spectra but due to not including the probably substantially larger model uncertainties. This creates the impression that the lifetimes and ODPs estimated here are far superior to previous work; which they might be, but this is currently not proven.

Author Response: We appreciate the reviewers' suggestion. However, such a comparison is beyond the scope of the present paper. We have clarified the text to emphasize that we are only addressing the uncertainty in the kinetic and photochemical data in the present paper. Considering uncertainty from other processes is beyond the scope of this paper.

Action Taken: See response to Referee #1s general comment for revisions to the text.

Referee Comment: L256-257 The term "fractional release factors" is explained nowhere in this manuscript. Why would they make a difference to the ODPs?

Author Response: We have added text to define this term, and provided context as to why this is important.

Action Taken: We have added the following text and appropriate references to the end of section 4.1: "The semi-empirical ODPs are dependent on observationally-based fractional release factors for a given stratospheric mean age of air, i.e., the fractional amount of a CFC that has been dissociated at a given point in the stratosphere (and the subsequent release of inorganic chlorine), relative to the amount of a CFC that entered at the tropopause (e.g. Schauffler et al., 2003; Newman et al., 2007; Daniel et al., 2007; Douglass et al., 2008; Laube et al., 2013). Differences in the semi-empirical vs. model ODPs in Table 6 are due, at least in part, to differences in the observationally based

fractional release factors taken for mid-latitude conditions compared to the global model calculations. Differences in the ODPs may also arise from differences in the Ko et al. (2013) lifetimes used for the semi-empirical ODPs vs. the model lifetimes, although these lifetime differences are small."

Referee Comment: L258-266 Again, the IR spectra derived in this work are not discussed at all. In this case it is not even mentioned that they were used in this calculation. Also, a comparison with published REs and GWPs would be useful here as well as in Table 6.

Author Response: We should acknowledge that we used our infrared spectra in the calculation. We should also have acknowledged the previous values for CFC-113a reported in the Etminan et al. (2014) study.

Action Taken: Text revised as follows: "Table 6 summarizes the radiative efficiencies (REs) for the CFCs calculated using the methods described in Hodnebrog et al. (2013) and the global warming potentials (GWPs) for the 20, 100, and 500-year time-horizons using the lifetimes and infrared spectra from this work. ".

The flowing sentence has been added: "Etminan et al. (2014) reported a RE of 0.23 W m-2 ppb-1 for CFC-113a and a GWP100 of 3310 using a lifetime of 51 years. These values are in reasonable agreement with the present results."